# Anony*mice*d Shareable Data: Using *mice* to Create and Analyze Multiply Imputed Synthetic Datasets

**Thom Benjamin Volker** *,† , **Gerko Vink** † 

Department of Methodology and Statistics, Utrecht University, Padualaan 14, 3584 CH Utrecht, The Netherlands; g.vink@uu.nl
* Correspondence: t.b.volker@uu.nl
† These authors contributed equally to this work.

**Abstract:** Synthetic datasets simultaneously allow for the dissemination of research data while protecting the privacy and confidentiality of respondents. Generating and analyzing synthetic datasets is straightforward, yet, a synthetic data analysis pipeline is seldom adopted by applied researchers. We outline a simple procedure for generating and analyzing synthetic datasets with the multiple imputation software `mice` (`Version 3.13.15`) in `R`. We demonstrate through simulations that the analysis results obtained on synthetic data yield unbiased and valid inferences and lead to synthetic records that cannot be distinguished from the true data records. The ease of use when synthesizing data with `mice` along with the validity of inferences obtained through this procedure opens up a wealth of possibilities for data dissemination and further research on initially private data.

**Keywords:** mice; multiple imputation; synthetic data; statistical disclosure control; privacy

## 1. Introduction

Open science, including open data, has been marked as the future of science [1], and the advantages of publicly available research data are numerous [2,3]. Collecting research data requires an enormous investment both in terms of time and monetary resources. Disseminating the research data increases the potential scientific returns for the same data collection effort tremendously. Additionally, the fact that public funds are used for data collection results in increasing demand for access to the collected data. Nevertheless, the possibilities for distribution of research data directly are often very limited due to restrictions of data privacy and data confidentiality. Although these regulations are much needed, privacy constraints are also ranked among the toughest challenges to overcome in the advancement of modern social science research [4].

Anonymizing research data might seem a quick and appealing approach to limit the unique identification of participants. However, this approach is not sufficient to fulfill contemporary privacy and confidentiality requirements [5,6]. Over the years, several other techniques have been used to increase the confidentiality of research data, such as categorizing continuous variables, top coding values above an upper bound, or adding random noise to the observed values [7]. However, these methods may distort the true data relation between variables, thereby reducing the data quality and the scientific returns for re-using the same data for further research.

An alternative solution has been proposed separately by Rubin [8] and Little [9]. Although their approaches differ to some extent, the overarching procedure is to use bonafide observed data to generate multiply imputed synthetic datasets that can be freely disclosed. While in practice, one could see this as replacing the observed data values by multiple draws from the predictive distribution of the observed data, based on some imputation model, Rubin would argue that these synthetic data values are merely drawn from the same true data generating model. In that sense, the observed data are never replaced, but the population is resampled from the information captured in the (incomplete)

sample. Using this approach, the researcher could replace the observed dataset as a whole with multiple synthetic versions. Alternatively, the researcher could opt to only replace a subset of the observed data. For example, one can choose to only replace dimensions in the data that could be compared with publicly available datasets or registers. Likewise, synthesization could be limited to those values that are disclosive, such as high incomes or high turnovers.

Conceptually, the synthetic data framework is based upon the building blocks of multiple imputation of missing data, as proposed by Rubin [10]. Instead of replacing just the missing values, one could easily replace any observed sensitive values by multiple draws from the posterior predictive distribution or a non-Bayesian predictive distribution (both of these will be referred to as predictive distributions throughout). Similar to multiple imputation of missing data, the multiple synthetic datasets allow for correct statistical inferences, despite the fact that the analyses do not use the *true* value. The analyses over multiple synthetic datasets should be pooled into a single inference, so that the researcher can draw valid conclusions from the pooled results. In that respect, the variance should reflect the added variability that is induced by the imputation procedure.

Potentially, this approach could fulfill the needs for openly accessible data, without running into barriers with regard to privacy and confidentiality constraints. However, there is no such thing as a free lunch: data collectors have to put effort in creating high-quality synthetic data. The quality of the synthetic data is highly dependent on the imputation models, and using flawed imputation models might bias subsequent analyses [11–13]. Conversely, if the models used to synthesize the data are able to preserve the relationships between the variables as in the original data, the synthetic data can be nearly as informative as the observed data. To fully exploit the benefits of synthetic data, complicating aspects of creating high-quality synthetic datasets should be kept at a minimum, while keeping the accuracy of the models as high as possible.

To mitigate unnecessary challenges related to creating synthetic data sets on behalf of the researcher, software aimed at multiple imputation of missing data can be employed. In particular, if researchers acquired familiarity with this software during earlier projects, or used it earlier in the research process, the additional burden of creating synthetic datasets is relatively small. The R-package `mice` [14] implements multiple imputation of missing data in a straightforward and user-friendly manner. However, the functionality of `mice` is not restricted to the imputation of missing data, but allows imputation of any value in the data, even observed values. Consequently, `mice` can be used for the creation of multiply imputed synthetic data sets.

After creating multiply imputed synthetic datasets, the goal is to obtain valid statistical inferences in the spirit of Rubin [10] and Neyman [15]. In the missing data framework, this is done by performing statistical analyses on all imputed datasets, and pooling the results of the analyses according to Rubin's rules ([10], p. 76). In the synthetic data framework, the same procedure is followed, but with a slight twist; there may be no values that remain constant over the synthetic datasets. The procedure of drawing valid inferences from multiple synthetic data sets is therefore slightly different.

In this manuscript, we detail a workflow for synthesizing data with `mice`. First, the `mice` algorithm for the creation of synthetic data will be briefly explained. The aim is to generate synthetic sets that reassure the privacy and confidentiality of the participants. Second, a straightforward workflow for imputation of synthetic data with `mice` will be demonstrated. Third, we demonstrate the validity of the procedure through statistical simulation.

## 2. Generating Synthetic Data with `mice`

Generally, there are three ways to impute data: joint modeling, sequential modeling, and fully conditional specification [12,16,17]. With joint modeling, the entire joint distribution of the data is specified as a single multivariate distribution, and the imputations are drawn from this model. This strategy is implemented in, among others, the R-package `jomo` [18] to deal with missing data. However, when the structure of the data increases

in complexity, specifying a single multivariate distribution that would fit the observed data can become challenging. A more flexible solution has been proposed in sequential regression imputation, in which the multivariate distribution is separated into univariate conditional distributions. Every single variable is then imputed conditionally on a model containing only the variables in the sequence located before the variable to be imputed. This approach has been implemented in the R-packages mdmb [19] for imputation of missing data and synthpop [20] for imputation of synthetic data.

Fully conditional specification (FCS) has been implemented in the mice package [14,21] in R [22], which has been developed for multiple imputation to overcome problems related to nonresponse. In that context, the aim is to replace missing values with plausible values from the predictive distribution of that variable. This is achieved by breaking down the multivariate distribution of the data $\mathbf{Y} = (\mathbf{Y}_{obs}, \mathbf{Y}_{mis})$ into $j = 1, 2, \ldots, k$ univariate conditional densities, where $k$ denotes the number of columns in the data. Using FCS, a model is constructed for every incomplete variable and the missing values $Y_{j,mis}$ are then imputed with draws from the predictive distribution of $P(Y_{j,mis}|\mathbf{Y}_{obs}, \theta)$ on a variable-by-variable basis. Note that the predictor matrix $Y_{-j}$ may contain yet imputed values from an earlier imputation step, and thus will be updated after every iteration. This procedure is applied $m$ times, resulting in $m$ completed datasets $\mathbf{D} = (\mathbf{D}^{(1)}, \mathbf{D}^{(2)}, \ldots, \mathbf{D}^{(m)})$, with $\mathbf{D}^{(l)} = (\mathbf{Y}_{obs}, \mathbf{Y}_{mis}^{(l)})$.

In mice (Version 3.13.15), the generation of multiply imputed datasets to solve for unobserved values is straightforward. The following pseudocode details multiple imputation of any dataset that contains missingness into the object imp with m = 10 imputated sets and maxit = 7 iterations for the algorithm to converge, using the default imputations methods.

```
library(mice)
imp <- mice(data.with.missings,
            m = 10,
            maxit = 7)
```

The imputation approach in mice can be easily extended to generate synthetic values. Rather than imputing missing data, observed values are then replaced by synthetic draws from the predictive distribution. For simplicity, assume that the data are completely observed (i.e., $\mathbf{Y} = \mathbf{Y}_{obs}$). Following the notation of Reiter and Raghunathan [23], let, given $n$ observations, $Z_i = 1$ if any of the values of unit $i = 1, 2, \ldots, n$, are to be replaced by imputations, and $Z_i = 0$ otherwise, with $Z = (Z_1, Z_2, \ldots, Z_n)$. Accordingly, the data consist of values that are to be replaced and values that are to be kept (i.e., $\mathbf{Y} = (\mathbf{Y}_{rep}, \mathbf{Y}_{nrep})$). Now, instead of imputing $\mathbf{Y}_{mis}$ with draws from the predictive distribution of $P(Y_{j,mis}|\mathbf{Y}_{obs}, \theta)$ as in the missing data case, $\mathbf{Y}_{rep}$ is imputed from the posterior distribution of $P(Y_{j,rep}^{(l)}|\mathbf{Y}_{-j}, Z, \theta)$, where $l$ is an indicator for the synthetic dataset ($l = 1, 2, \ldots, m$). This process results in the synthetic data $\mathbf{D} = (\mathbf{D}^{(1)}, \mathbf{D}^{(2)}, \ldots, \mathbf{D}^{(m)})$.

Hence, creating synthetic values for a completely observed dataset into the object syn, given the same imputation parameters as specified above, can be realized as follows:

```
syn <- mice(data.to.synthesize,
            m = 10,
            maxit = 1,
            where = matrix(TRUE,
                           nrow = nrow(data.to.synthesize),
                           ncol = ncol(data.to.synthetize)))
```

Here, the argument where requires a matrix of the same dimensions as the data, (i.e., a $n \times k$ matrix) containing logicals $z_{ij}$ that indicate which cells are selected to have their values replaced by draws from the predictive distribution. This approach enables replacing a subset of the observed data (e.g., by specifying only those cells that are to be replaced as

TRUE in the `where`-matrix, leaving the rest to FALSE), or as in the aforementioned example, the observed data as a whole, resulting in a dataset that partially or completely consists of synthetic data values. Note that because the data are completely observed, iterating over the predictive distribution is not required.

If the goal is to create synthetic versions of a dataset that suffers from missingness, the imputation procedure becomes more complex. A two-step imputation procedure has been proposed by Reiter [11]. In the first step, the missing data are imputed according to the principles of multiple imputation for missing data, while in a second step, the synthetic data are imputed to conform the corresponding procedure. Ideally, these steps could be combined using `mice`, conditioning the imputations on both the missingness and the values that are to be synthesized. In practice, replacing observed and missing cells in a dataset by synthetic values into the object `syn.mis`, given the same imputation parameters as before, can be realized by the following code execution.

```
syn.mis <- mice(data.with.missings.to.synthesize,
                m = 10,
                maxit = 7,
                where = matrix(TRUE,
                               nrow = nrow(data.with.missings.to.synthesize),
                               ncol = ncol(data.with.missings.to.synthesize)))
```

However, because the validity of this approach when the data suffer from missingness has not been investigated yet, we restrict ourselves in the following sections to a completely observed data set.

Choosing an adequate imputation model to impute the data is paramount, as a flawed imputation model may drastically impact the validity of inferences [12,13]. Imputation models should be as flexible as possible to capture most of the patterns in the data, and to model possibly unanticipated data characteristics [16,24]. Generally, multiple methods can be feasible, including joint modeling, sequential regression and FCS, all with different parameter specifications, depending on the imputation problem at hand [17,25,26]. Various simulations have shown that the most accurate imputation procedure may vary over different imputation problems (e.g., see [27] for an overview). Hence, a one-size-fits-all solution does probably not exist.

However, one of the methods that has shown a reasonable and consistent performance over multiple simulations (e.g., [28–31]) is FCS in combination with classification and regression trees (CART; [32]). Due to its nonparametric nature, CART enables modeling complex patterns (e.g., interactions and nonlinear relationships) in the data. Loosely speaking, CART sequentially splits the predictor space into non-overlapping regions in such a way that the within-region variance is as small as possible after every split. As such, CART does not impose any parametric distribution on the data, making it a widely applicable method that allows for a large variety of relationships within the data to be modeled [33]. Given these appealing characteristics and the call for the use of flexible methods when multiply imputing data, we will focus our illustrations and evaluations of `mice` to method `mice.impute.cart()`, realized by:

```
syn <- mice(data.to.synthesize,
            m = 10,
            maxit = 1,
            method = "cart",
            where = matrix(TRUE,
                           nrow = nrow(boys),
                           ncol = ncol(boys)))
```

In a nutshell, the above code shows the simplicity of creating $m = 10$ synthetic datasets using `mice`. In practice, however, one should take some additional complicating factors into account. For example, one should account for deterministic relations in the data.

Additionally, relations between variables may be described best using a different model than `CART`. Such factors are data dependent and should be considered by the imputer. In the next section, we will describe how a completely observed version of the `mice::boys` data [34] can be adequately synthesized. Additionally, we will show through simulations that this approach yields valid inferences.

### 3. Materials and Methods

We demonstrate the suitability of using `mice` for synthesization using a simulation study on the `mice::boys` dataset. This data set consists of the values of 748 Dutch boys on nine variables (Table 1).

**Table 1.** Description of the features in the `mice::boys` data set.

| Column | Description |
| --- | --- |
| age | age in years |
| hgt | height (cm) |
| wgt | weight (kg) |
| bmi | body mass index |
| hc | head circumference (cm) |
| gen | genital Tanner stage G1–G5 |
| phb | pubic hair Tanner P1–P6 |
| tv | testicular volume (mL) |
| reg | region |

Unfortunately, this dataset does not differ from the vast majority of collected datasets, in the sense that it suffers from missing data. For simplicity, we solve the missingness problem using a single imputation with the default `mice` imputation model for all predictors except `bmi`, which is passively imputed using its deterministic relation with `wgt` and `hgt`. Specifically, the imputed values are used to calculate the exact `bmi` values that correspond with `hgt` and `wgt`.

*Simulation Methods*

Usually, one would draw samples from a population that can be synthesized to evaluate the performance of the synthesization methods. As we only have access to a single sample, 1000 bootstrap samples of the `boys` data have been synthesized with $m = 5$ imputations for every data cell to induce an appropriate amount of sampling variance. This approach precludes us from knowing the *true* data-generating model, and in this sense provides a more stringent test of the applicability of `mice` for real-life synthesization procedures. Specifically, every bootstrapped sample is treated as an actual sample from a population. These bootstrap samples will be synthesized using `mice`, using a model that is built to approximate the true data-generating mechanisms as closely as possible. As relationships in realistic data are generally more complex than relationships in parametrically simulated data, achieving good performance on a real data set is likely to be more indicative of practical applicability than good performance on samples from a known multivariate probability density function.

In the simulations, synthetic values are generated using the `CART` imputation method for all columns, except for `bmi`. The deterministic relation `bmi` which will be synthesized passively based on the synthetic values for `hgt` and `wgt` to preserve the relation in the synthetic data. Additional parameters that come with the use of `mice.impute.cart()` are the complexity parameter `cp` and the minimum number of observations in any terminal node `minbucket`, that both constrain the flexibility of the imputation model. The values of the parameters `cp` and `minbucket` ought to adhere to the call for imputation models that are as flexible as possible. Appropriate values for these parameters, as well as the input for the `predictorMatrix`, depend on the data at hand. In the current example, the complexity parameter is specified at `cp` = $10^{-8}$ rather than the default value $10^{-4}$, and the minimum

number of observations in each terminal node is set at `minbucket = 3` rather than the default value 5. By allowing for more complexity in the imputation model, bias in the estimates from the synthetic dataset is reduced. Additionally, since the synthesis pattern is monotone, the number of iterations can be set to `maxit = 1` (e.g., [7], Ch. 3).

To assess the performance of `mice` for synthesizing data, we compare the bootstrapped samples with the synthetic versions of these bootstrapped samples. Specifically, univariate descriptive statistics, the correlation matrix, and two linear regression models as well as one ordered logistic regression model will be considered. Subsequently, the bias in the parameters and the 95% confidence interval coverage of the synthetic data will be examined. Similar to multiple imputation of missing data, correct inferences from synthetic data require correct pooling over the multiply imputed data sets.

Obtaining a final point estimate of the parameter of interest $Q$ after imputation is fairly easy and no different from pooling in the case of missing data [10]. One can calculate the average of the $m$ point estimates $q^{(l)}$

$$\bar{q}_m = \sum_{l=1}^{m} \frac{q^{(l)}}{m},$$

with $l = 1, \ldots, m$.

Furthermore, similarly to the missing data case, variances, and subsequently confidence intervals, should incorporate the increase in variance that is due to imputation [7,35]. Yet, the increase in variance due to imputation differs according to whether missing values are imputed or observed data are replaced by synthetic values. Whereas the variance estimate after imputation of missing data needs to account for the fact that a certain amount of information in the data is missing, variance estimation from synthetic data does not suffer from this issue. The adjusted variance estimate that follows from using multiple synthetic datasets only suffers from the fact that a finite number of $m$ synthetic datasets are used to resemble the observed data. Hence, the according variance estimate for synthetic data as developed by Reiter [35] yields

$$T = \bar{u}_m + \frac{b_m}{m},$$

with between-imputation variance

$$b_m = \sum_{l=1}^{m} \frac{(q^{(l)} - \bar{q}_m)^2}{(m-1)},$$

and sampling variance

$$\bar{u}_m = \sum_{l=1}^{m} \frac{u^{(l)}}{m},$$

where $u^{(l)}$ denotes the variance estimate in the $l$th synthetic dataset. These pooling rules are implemented in `mice` as the function `pool.syn()`. Returning to our previous example, pooling the parameter estimates of a linear model in which the dependent variable `DV` is regressed on predictors `IV1` and `IV2` fitted on each of the synthetic datasets is as straightforward as the following code block.

```
fits <- with(syn, lm(DV ~ IV1 + IV2))
pool.syn(fits)
```

Reiter and Kinney [36] proved that synthesizing data from the posterior predictive distribution is not required for these pooling rules to hold. In fact, the parameters of an imputation model can be fixed to their maximum likelihood estimates or their posterior modes, and synthetic values can be drawn after plugging these values into the imputation models. This observation has been taken a step further by Raab et al. [31], who developed

pooling rules that do not require multiple synthetic datasets and performed highly similar to the aforementioned pooling rules. However, because these pooling rules can only be used when the complete data (rather than a subset of the data) are synthesized, we adhere to the originally proposed pooling rules by Reiter [35], who defined the total variance in terms of sampling variance and between-imputation variance.

## 4. Results

The synthetic data are evaluated with respect to the *true* dataset on the basis of three aims. We believe that every reliable and valid data synthesization effort in statistical data analysis should be able to yield (1) unbiased univariate statistical properties, (2) unbiased bivariate properties, (3) unbiased and valid multivariate inferences, and (4) synthetic data that cannot be distinguished from real data. We consider the evaluation of the synthetic data simulations in the above order.

### 4.1. Univariate Estimates

The univariate descriptives for the original data and the synthetic data can be found in Table 2. From this table, it can be seen that the synthetic data estimates closely resemble the true data estimates. All sample statistics of interest show negligible bias over the 1000 synthetic data sets.

**Table 2.** Univariate descriptives for the true data and $m = 5$ pooled univariate descriptives for the synthetic data over 1000 simulations. Variable names followed by an asterix (*) are categorical.

|  | n | Mean | sd | Median | Min | Max | Skew | Kurtosis |
|---|---|---|---|---|---|---|---|---|
| original age | 748 | 9.16 | 6.89 | 10.50 | 0.04 | 21.18 | −0.03 | −1.56 |
| synthetic age | 748 | 9.15 | 6.89 | 10.49 | 0.04 | 20.96 | −0.03 | −1.55 |
| original hgt | 748 | 131.10 | 46.52 | 145.75 | 50.00 | 198.00 | −0.30 | −1.47 |
| synthetic hgt | 748 | 131.06 | 46.50 | 145.32 | 50.69 | 197.16 | −0.30 | −1.47 |
| original wgt | 748 | 37.12 | 26.03 | 34.55 | 3.14 | 117.40 | 0.38 | −1.03 |
| synthetic wgt | 748 | 37.09 | 26.00 | 34.44 | 3.35 | 112.26 | 0.38 | −1.03 |
| original bmi | 748 | 18.04 | 3.04 | 17.45 | 11.73 | 31.74 | 1.14 | 1.79 |
| synthetic bmi | 748 | 18.05 | 3.08 | 17.48 | 11.49 | 32.37 | 1.11 | 1.85 |
| original hc | 748 | 51.62 | 5.86 | 53.10 | 33.70 | 65.00 | −0.91 | 0.12 |
| synthetic hc | 748 | 51.61 | 5.86 | 53.18 | 34.38 | 62.85 | −0.91 | 0.12 |
| original gen * | 748 | 2.53 | 1.59 | 2.00 | 1.00 | 5.00 | 0.52 | −1.36 |
| synthetic gen * | 748 | 2.53 | 1.59 | 2.00 | 1.00 | 5.00 | 0.52 | −1.35 |
| original phb * | 748 | 2.75 | 1.86 | 2.00 | 1.00 | 6.00 | 0.56 | −1.25 |
| synthetic phb * | 748 | 2.75 | 1.86 | 2.00 | 1.00 | 6.00 | 0.56 | −1.24 |
| original tv | 748 | 8.43 | 8.12 | 3.00 | 1.00 | 25.00 | 0.85 | −0.78 |
| synthetic tv | 748 | 8.42 | 8.11 | 3.19 | 1.00 | 25.00 | 0.85 | −0.77 |
| original reg * | 748 | 3.02 | 1.14 | 3.00 | 1.00 | 5.00 | −0.08 | −0.77 |
| synthetic reg * | 748 | 3.02 | 1.14 | 3.00 | 1.00 | 5.00 | −0.08 | −0.76 |

The sampling distribution of the means of the continuous variables for both the bootstrapped and the synthetic data can be found in Figure 1, while the number of observations per category in the bootstrapped data and the averaged number of observations per category over the synthetic datasets can be found in Figure 2.

Both figures show that the marginal distributions of the bootstrapped data and the synthetic data are highly similar. The sampling distributions of these parameters hardly seem to differ. Hence, both the numeric and graphical results show that univariately the synthesis model proves adequate.

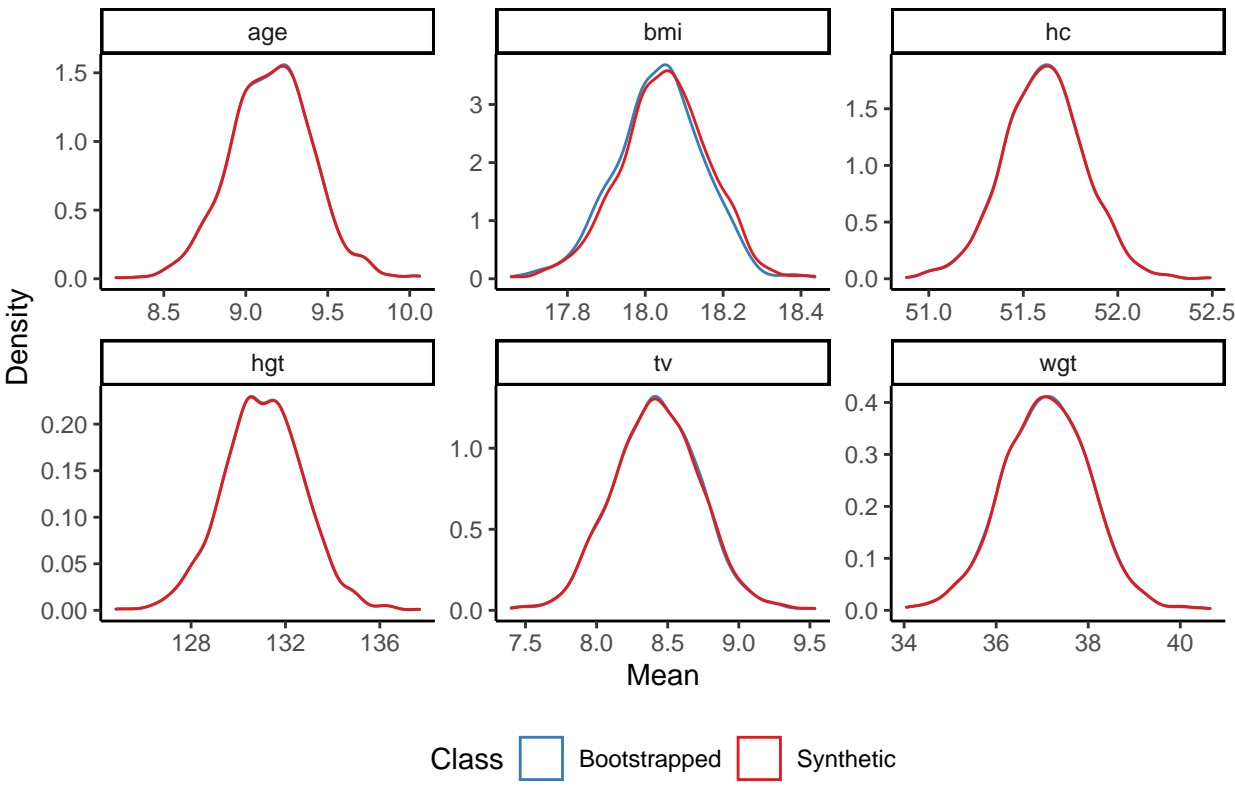

**Figure 1.** Empirical sampling distribution of the bootstrapped and synthetic means of the continuous variables in the boys data.

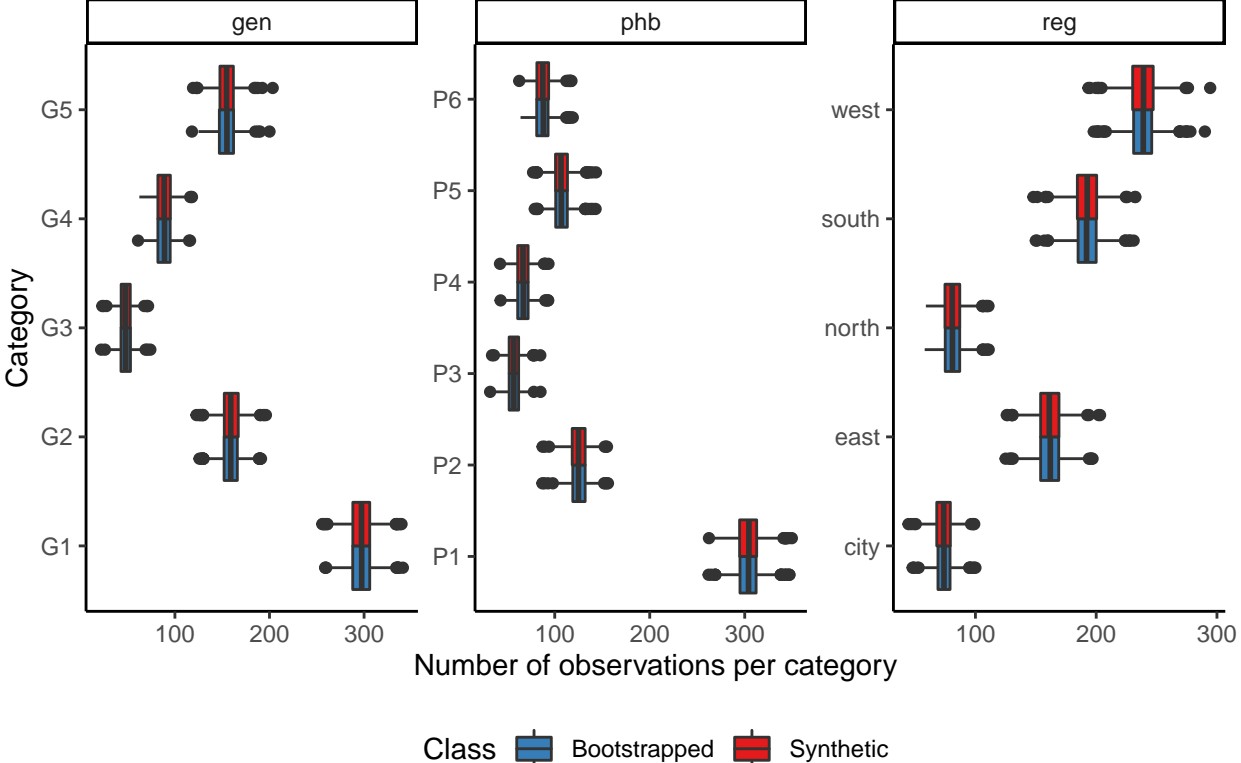

**Figure 2.** Empirical sampling distribution of the number of observations within each category of the categorical variables in the bootstrapped and synthetic boys data.

## 4.2. Bivariate Estimates

An often used bivariate statistic is Pearson's correlation coefficient. When evaluating this correlation coefficient on the numeric columns in the boys dataset, the differences between the correlations in the synthetic and bootstrapped data are very small. These results are displayed in Table 3.

**Table 3.** Bivariate correlations of the numerical columns in the true data with in parentheses the corresponding bias of the $m = 5$ pooled synthetic correlations over 1000 simulations. All estimates are rounded to 3 decimal places.

|      | Age           | hgt           | wgt           | bmi           | hc            | tv            |
|------|---------------|---------------|---------------|---------------|---------------|---------------|
| age  | 1             | 0.976 (0.001) | 0.950 (0.000) | 0.627 (0.009) | 0.853 (0.000) | 0.810 (0.002) |
| hgt  | 0.976 (0.001) | 1             | 0.944 (0.001) | 0.596 (0.013) | 0.907 (0.000) | 0.754 (0.000) |
| wgt  | 0.950 (0.000) | 0.944 (0.001) | 1             | 0.791 (0.009) | 0.834 (0.000) | 0.817 (0.000) |
| bmi  | 0.627 (0.009) | 0.596 (0.013) | 0.791 (0.009) | 1             | 0.588 (0.009) | 0.610 (0.007) |
| hc   | 0.853 (0.000) | 0.907 (0.000) | 0.834 (0.000) | 0.588 (0.009) | 1             | 0.623 (0.000) |
| tv   | 0.810 (0.002) | 0.754 (0.000) | 0.817 (0.000) | 0.610 (0.007) | 0.623 (0.000) | 1             |

The correlations obtained from synthetic data are unbiased with respect to the true dataset. The largest bias over 1000 simulations equals 0.013, indicating that the imputation model is capable of preserving the bivariate relations in the data.

## 4.3. Multivariate Model Inferences

First, we evaluate the performance of our synthetic simulation set on a linear model where hgt is modeled by a continuous predictor age and an ordered categorical predictor phb. The results for this simulation can be found in Table 4.

**Table 4.** Simulation results for a linear regression model with continuous and ordered categorical predictors. The model evaluated is hgt $\sim$ age + phb. Depicted are the true data estimates and the bias from the true data estimates and the coverage rate of the 95% confidence interval for the bootstrap and synthetic data sets.

|             |          | Bootstrap |       | Synthetic |       |
|-------------|----------|-----------|-------|-----------|-------|
| Term        | Estimate | Bias      | Cov   | Bias      | Cov   |
| (Intercept) | 63.087   | −0.001    | 0.970 | 0.405     | 0.958 |
| age         | 7.174    | 0.000     | 0.958 | −0.033    | 0.947 |
| phb.L       | −12.250  | 0.008     | 0.950 | 0.582     | 0.927 |
| phb.Q       | −1.376   | −0.022    | 0.926 | 0.112     | 0.934 |
| phb.C       | −3.564   | 0.051     | 0.915 | 0.301     | 0.912 |
| phb^4       | −0.431   | 0.016     | 0.930 | 0.106     | 0.940 |
| phb^5       | 2.064    | 0.060     | 0.941 | 0.077     | 0.943 |

We see that the finite nature of the true dataset together with the design-based simulation setup yields slight undercoverage for the terms of phb. This finding is observed in both the bootstrap coverages (i.e., the fraction of 95% confidence intervals that cover the true data parameters) and the synthetic data coverages. Hence, it is likely that this undercoverage stems from the simulation setup, rather than the imputation procedure. Besides the undercoverage, there is a tiny bit of bias in the estimated coefficients of the variable phb that occurs in the synthetic estimates, but not in the observed estimates. Yet, since the bias is relatively small and does not result in confidence invalidity, it seems fair to assume that the introduced bias is not that problematic.

Second, we evaluate a proportional odds logistic regression model in which the ordered categorical variable gen is modeled by continuous predictors age and hc, and the categorical predictor reg. The results for this model are shown in Table 5.

**Table 5.** Simulation results for a proportional odds logistic regression model with continuous and ordered categorical predictors. The model evaluated is `gen ~ age + hc + reg`. Depicted are the true data estimates and the bias from the true data estimates and the coverage rate of the 95% confidence interval for the bootstrap and synthetic data sets.

| Term | Estimate | Bootstrap | | Synthetic | |
|---|---|---|---|---|---|
| | | Bias | Cov | Bias | Cov |
| age | 0.461 | 0.004 | 0.942 | 0.002 | 0.939 |
| hc | −0.188 | −0.000 | 0.929 | −0.004 | 0.945 |
| regeast | −0.339 | 0.012 | 0.960 | 0.092 | 0.957 |
| regwest | 0.486 | 0.009 | 0.952 | −0.122 | 0.944 |
| regsouth | 0.646 | 0.012 | 0.966 | −0.152 | 0.943 |
| regcity | −0.069 | 0.012 | 0.940 | 0.001 | 0.972 |
| G1\|G2 | −6.322 | 0.032 | 0.934 | −0.254 | 0.946 |
| G2\|G3 | −4.501 | 0.052 | 0.936 | −0.246 | 0.945 |
| G3\|G4 | −3.842 | 0.058 | 0.937 | −0.244 | 0.948 |
| G4\|G5 | −2.639 | 0.064 | 0.936 | −0.253 | 0.947 |

These results demonstrate that the synthetic data analysis yields inferences that are on par with inferences from the analyses directly on the bootstrapped datasets. Hence, for the regression coefficients as well as the intercepts, the analyses on the synthetic data yield valid results. Nevertheless, a small amount of bias is introduced in the estimated intercepts of the synthetic data. However, the corresponding confidence interval coverage rates are actually somewhat higher than the confidence interval coverage rates of the bootstrapped data. Therefore, the corresponding inferences do not seem to be affected by this small bias.

### 4.4. Data Discrimination

When we combine the original and synthetic data, can we predict which rows come from the synthetic dataset? If so, then our synthetic data procedure would be redundant, since the synthetic set differs from the observed set. To evaluate whether we can distinguish between the true data and the synthetic data, we combine the rows from each simulation synthetic dataset with the rows from the true data. We then run a logistic regression model to predict group membership: i.e., does a row belong to the true data or synthetic data. As predictors we take all columns in the data. The pooled parameter estimates over all simulations can be found in Table 6.

**Table 6.** Simulation results for a logistic regression model aimed at discriminating between synthetic records and true records.

| Term | Estimate | std Error | Statistic | df | p Value |
|---|---|---|---|---|---|
| (Intercept) | 0.22 | 1.15 | 0.19 | 521.93 | 0.60 |
| wgt | 0.00 | 0.02 | 0.25 | 420.19 | 0.60 |
| hgt | −0.00 | 0.01 | −0.18 | 359.73 | 0.60 |
| age | −0.00 | 0.05 | −0.00 | 345.73 | 0.62 |
| hc | 0.00 | 0.03 | 0.11 | 415.44 | 0.61 |
| gen.L | −0.00 | 0.42 | −0.00 | 164.76 | 0.65 |
| gen.Q | −0.01 | 0.21 | −0.04 | 203.38 | 0.63 |
| gen.C | −0.00 | 0.16 | −0.02 | 239.63 | 0.64 |
| gen^4 | 0.01 | 0.21 | 0.01 | 237.41 | 0.63 |
| phb.L | −0.02 | 0.44 | −0.04 | 156.54 | 0.64 |
| phb.Q | −0.01 | 0.22 | −0.02 | 198.20 | 0.64 |
| phb.C | 0.00 | 0.18 | 0.00 | 211.17 | 0.62 |
| phb^4 | 0.00 | 0.18 | 0.02 | 228.51 | 0.64 |
| phb^5 | 0.00 | 0.20 | 0.02 | 248.57 | 0.63 |
| tv | 0.00 | 0.02 | 0.01 | 264.26 | 0.63 |
| regeast | −0.00 | 0.23 | 0.01 | 210.90 | 0.64 |
| regwest | −0.00 | 0.22 | −0.00 | 221.14 | 0.63 |
| regsouth | −0.01 | 0.22 | −0.02 | 226.10 | 0.64 |
| regcity | 0.01 | 0.27 | 0.03 | 204.15 | 0.64 |
| bmi | −0.02 | 0.06 | −0.26 | 320.18 | 0.61 |

From these pooled results we can see that the effects for all predictors are close to zero and nonsignificant. When we take the predicted values from the simulated models and compare them with the *real* values, we obtain the summary statistics in Table 7.

**Table 7.** Confusion statistics for a prediction model aimed at discriminating between synthetic records and true records.

|  | Estimate |
| --- | --- |
| Accuracy | 0.50381 |
| Balanced Accuracy | 0.50381 |
| Kappa | 0.00762 |
| McNemar *p* Value | 0.63187 |
| Sensitivity | 0.50368 |
| Specificity | 0.50394 |
| Prevalence | 0.50000 |

The accuracy of the predictive modeling effort is not better than random selection, and the Kappa coefficient indicates that a perfect prediction model is far from realized. The accuracy is quite balanced as there is no skewness over sensitivity and specificity. These findings indicate that the synthetic data are indistinguishable from the true data.

## 5. Discussion

We demonstrated that generating synthetic datasets with `mice` using CART in `R` is a straightforward process that fits well in a data analysis pipeline. The approach is hardly different from using multiple imputation to solve problems related to missing data and, hence, can be expected to be familiar to applied researchers. The multiple synthetic sets yield valid inferences on the true underlying data generating mechanism, thereby capturing the nature of the original data. This makes the multiple synthetisation procedure with `mice` suitable for further dissemination of synthetic datasets. It is important to note that in our simulations we used a single iteration and relied on CART as the imputation method. A single iteration is sufficient only when the true data are completely observed, or when the missingness pattern is monotone [7]. If both observed and unobserved values are to be synthesized, then more iterations and a careful investigation into the convergence of the algorithm are in order. Synthetic data generation is therein no different than multiple imputation.

When creating synthetic data with `mice`, close attention should be paid to three distinct factors: (i) the additional uncertainty that is due to synthesizing (part of) the data should be incorporated, (ii) potential disclosure risks that remain after synthesizing the data should be assessed, and (iii) the utility of the synthetic data should be examined. First, the procedure of generating multiple synthetic sets may seem overly complicated. We would like to emphasize that analyzing a single synthesized set, while perhaps unbiased, would underestimate the variance properties that are so important in drawing statistical inferences from finite datasets. After all, we are often not interested in the sample at hand, but aim to make inferences about the underlying data generating mechanism as reflected in the population. Properly capturing the uncertainty of synthetic datasets, just like with incomplete datasets, is therefore paramount. To achieve this, we adopted a bootstrap scheme in our simulations, to represent sampling from a superpopulation. However, when a single sample should be the reference, in the sense that the sample reflects the population one wishes to make inferences about, adjusted pooling rules are required that are similar to the procedure outlined in Vink and van Buuren [37]. The corresponding pooling rules have not been derived yet and the incorporation in data analysis workflows would require proper attention from developers alike.

Second, it is important that imputers pay close attention to disclosure risks that remain after creating synthetic datasets. Creating synthetic data, unless generated from a completely parametric distribution, does not remove all potential disclosure risks. For example,

if the values that ought to be replaced have the exact same value imputed, the synthesization procedure has no use. Additionally, if not all variables in the data are synthesized, but the variables that are synthesized can be linked to open access data, a disclosure risk may remain [38]. If the open access data allow for identification, and the corresponding observations in the synthetic data can be identified, the variables that are not synthesized may provide information that should not have been disseminated. The associated problems generally decrease when the sensitive variables are synthesized as well. Still, it is important to remain critical regarding the extent to which the synthetic data might be disclosive. The practical development of easy-to-use software to identify which observations are at risk of disclosure is an area of future work that can improve these issues. Subsequently, the implementation of ways to overcome such problems once detected, for example by record swapping as suggested by Jiang et al. [13], is welcomed.

Third, the utility of the synthetic data should receive considerable scrutiny. After all, synthetic data are generated to allow third parties without access to the data to draw valid inferences. If the quality of the synthetic data is very poor, the validity of inferences is at stake. We showed that CART proved to be an adequate method for the data at hand. However, it may well be that for some problems, using CART for imputation is not the optimal strategy. We already demonstrated that imputing deterministic relationships should not be based on CART. However, due to the variety in possible imputation methods that have proved to perform well in varying circumstances, providing a single solution to all synthesis problems seems impossible. If required, imputers could build up their synthesis models in complexity. Nevertheless, similar to past research (e.g., [29–31]), we showed that CART provides adequate results in general, and therefore serves as a good starting point.

After imputation, the quality of the synthetic data can be assessed using various utility measures. The simplest way to assess data quality is through visual inspection. However, visualizing more than two dimensions quickly becomes cumbersome, which makes it hard to assess the accuracy of multivariate relations. Trying to predict whether the data are observed or synthetic enables an intuition of the quality of the synthesis. This procedure has been advanced by Snoke et al. [39], who proposed a measure of distributional similarity based on the mean squared error of propensity scores indicating class belonging for both observed and synthetic data. Additionally, if one knows the analyses that will be carried out by the synthetic data users, it is possible to compare the results on the observed and the synthetic data, enabling the assessment of the accuracy of the synthesization method. Both of these methods have been implemented in the R-package synthpop [20], who pioneered in the field of synthetic data generation in R.

While synthpop is capable of synthesizing regardless of missing data, it is not developed to solve missingness problems. If the data for synthesization contain missing values, the missingness will be considered as a property of the data, and the corresponding synthetic datasets will contain missing values as well. If the goal is to solve for the missingness and synthesize the dataset, a two-step approach must be used, in which the missing values are imputed using a different package, such as mice. Subsequently, synthpop can be used to synthesize the imputed data. Then, given $m$ multiple imputations and $r$ synthesizations, at least $m \times r$ synthetic datasets are in order, after which the data can be analysed and the analyses can be pooled using rules developed by Reiter [11]. However, such an approach would be computationally inefficient and practically cumbersome, because the architectures of both packages differ.

The appealing nature of mice therefore lays in its ability to both solve the missingness problem and create synthetic versions of the data. Eventually, the flexibility with mice is that unobserved and observed data values could be synthesized at once, without the need for a two-step approach. Using mice, $m$ synthetic sets would be sufficient. However, as of today, no pooling rules for one-step imputation of missingness and synthesization have been developed. The derivation of those would yield a massive theoretical improvement to further reduce the burden of creating synthetic data sets.

That said, the fields of synthetic data and multiple imputation are very much in development. Although efforts have been made to bridge gaps between these fields, such as in the `miceadds` package [40], which links functionality between between `synthpop` and `mice`, further improvements could be made in that respect. Specifically, `synthpop` contains a comprehensive suite of functions that can be used to assess the quality of synthetic data (e.g., see [39]), while `mice` contains methodology to allow for flexible synthesis of intricate data structures, such as multilevel data. Being able to use such functionality interchangeably would greatly benefit data analysts.

The added possibility of synthesizing data with `mice` in `R`, together with the validity of inferences obtained through this procedure opens up a wealth of possibilities for data dissemination and further research of initially private data to applied `R` users.

**Author Contributions:** Conceptualization, T.B.V. and G.V.; methodology, T.B.V. and G.V., software, T.B.V. and G.V.; analysis, T.B.V. and G.V.; writing manuscript, T.B.V. and G.V. All authors have read and agreed to the published version of the manuscript.

**Funding:** This research received no external funding.

**Institutional Review Board Statement:** Not applicable.

**Informed Consent Statement:** Not applicable.

**Data Availability Statement:** A full simulation archive is available from (GitHub https://github.com/amices/Synthemice/tree/master/manuscript).

**Acknowledgments:** We are grateful to Mirthe Hendriks and Stijn van den Broek for replicating the simulation study on an independent dataset. Additionally, we thank Hanne Oberman, two anonymous reviewers, and the editor for their helpful and insightful comments.

**Conflicts of Interest:** The authors declare no conflict of interest.

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
