# Peer review of "Anonymiced Shareable Data: Using mice to Create and Analyze Multiply Imputed Synthetic Datasets"

_psych, doi:10.3390/psych3040045_

Round 1

Reviewer 1 Report

I have signed this review because my identity will be obvious from below. 

I am one of the authors of the synthpop package that creates synthetic data in R. There is one reference to one of our papers in the submission (reference 30) but the point it makes is wrong and there is no reference to any of our other work. There is a lot that is sensible in the paper and part of the review of the literature is OK.  Here are details.

Point 1 Starting on line 263 the authors state

"For example, the synthpop[30] package in R would yield valid synthetic data sets, but requires the true data to be completelyobserved."

They cannot have ever used synthpop.  Missing values are dealt with as follows. For categorical data they are made into a seperate category. For continuous variables an indicator variable is formed that is then included in the synthesis sequence before the variable with missing values and then is available as a predictor of subsequent variables.  There is also an option for missing value codes additional to NA, such as 9999, to be treated as additional missing value codes and synthesised appropriately, At the end of the synthesis the new data set is created to match the original with synthesised missing values.

Point 2 Throughout the authors assume that synthesis requires multiple data sets to be created in the same way as is the case for imputation. As we show in our paper Raab G, Nowok B, Dibben “Practical data synthesis for large samples.” Journal of Privacy and Confidentiality, 7, 67–97. there are important differences between synthesis and imputation. This paper introduces variance estimates that do not require multiple data sets. It also shows that, unlike imputation, it is not necessary to generate samples from the posterior predictive distribution for synthesis. It also shows that these estmates perform much better than the one cited in the paper that requires multiple syntheses. See also the vignette on inference available as part of the synthpop package.

Point3 As well as the paper mentioned above the authors do not cite any of our work on the theory and practice of synthetic data, References can be found on the resources page of www.synthpop.org.uk.

Point 4 Their discussion of the evaluation of the utility of synthetic data is a good shot at it, but it reinvents a wheel that we and others have been using for many years. See the paper by Snoke et al. from the synthpop web cite. We are also in the process of expanding the utility functions available in synthpop and the newest version of the package, currently on github but soon to be uploaded to CRAN allows them to be used with data sets created by other methods. Our draft  paper reviewing utility methods for synthetic data can be accessed here https://arxiv.org/abs/2109.12717

Comment

There are clear parallels between imputation and synthesis in that both of them can use teh full conditional specification to define the distribution. We acknowledged our dept to the excellent MICE package in our papers describing the package. This paper is a good effort to use MICE for synthesis. It parallels another effort to do the same for IVEWARE for SAS. I think Ragunathan was one of the authors but i don't have the details to hand.  Both approaches have the defects mentioned above. More importantly they are lacking in many of the additional functionality that we have added to synthpop over the various versions we have developed over the years - see the news file on a recent versio of the package.

Author Response

Dear reviewer,

We greatly appreciate your review, and have tried to incorporate your feedback accordingly. Please see our detailed response in the attached file, and the revised version of the manuscript.

Sincerely, 

The authors

Reviewer 2 Report

The submitted paper illustrates how the popular R package mice can be used to create synthetic data sets. As pointed out by the authors (and many other scholars before), synthetic data can be a powerful tool to share and publish data that closely resemble the original data but at the same time protect the confidentiality of the participants (i.e., limit the possibility to identify individual participants). Using an example data set, the paper clearly explains the mice syntax and the close connection between multiply imputed data and synthetic data. I have a few comments for the authors to consider.

First, one crucial aspect of generating synthetic data is the specification of the synthesis model (i.e., imputation model in mice). It is argued that more flexible specifications should be preferred because they can pick up possible non-linear relationships that may be present in the data. However, there are many possible choices for a flexible specification of the imputation model and there is a huge literature about this topic. I doubt that the CART approach (that is proposed in the present paper) in combination with a fully conditional specification (FCS) should always be preferred. I think a more balanced discussion of other modeling approaches (e.g., joint modelling approaches with flexible parametric specifications) is needed.

Second, the logic of the simulation study needs to be better explained. Why did the authors use bootstrapping to generate samples from the observed data set? Usually, in simulation studies data are generated from a population model in order to evaluate the quality of different estimators. What is the target of inference in the presented simulations?

Third, I think it would be important (in section 2) to add some more technical details about how mice generates synthetic data in the presence of missing data – particularly when missing data pattern are not monotone.

Fourth, I would suggest to include a paragraph in the Discussion in which other software options are mentioned and discussed. The basic idea of the suggested approach in the paper is to overimpute data using a multiple imputation approach. This can also be implemented with other imputation software. Although, I agree with the authors that the popular mice software is particularly convenient. In addition, the synthpop package provides many helpful tools for creating and analyzing synthetic data. This should also be emphasized.

Fifth, it needs to be clearly mentioned that the proposed approach and subsequent analyses with synthetic data strongly depend on the correct specification of the synthesis model. If the synthesis model does not capture relationships among variables (e.g., interaction effects, non-linear effects) that are of interest in the main analysis, the synthetic data will produce biased estimates. I think it is important that applied researchers are aware of this issue when they publish/share their synthetic data.

Author Response

(The authors gave the same response as above.)
